# Definition and Quantification of Three-Dimensional Imaging Targets to Phenotype Pre-Eclampsia Subtypes: An Exploratory Study

**DOI:** 10.3390/ijms24043240

**Published:** 2023-02-07

**Authors:** Sammy Hermans, Jacob Pilon, Dennis Eschweiler, Johannes Stegmaier, Carmen A. H. Severens–Rijvers, Salwan Al-Nasiry, Marc van Zandvoort, Dimitrios Kapsokalyvas

**Affiliations:** 1Department of Genetics and Cell Biology, Maastricht University, 6200 MD Maastricht, The Netherlands; 2Institute of Imaging and Computer Vision, RWTH Aachen University, 52074 Aachen, Germany; 3Pathology, GROW, Maastricht University Medical Centre (MUMC), 6229 HX Maastricht, The Netherlands; 4Obstetrics and Gynaecology, GROW, Maastricht University Medical Centre (MUMC), 6229 HX Maastricht, The Netherlands; 5Department of Genetics and Cell Biology, GROW, CARIM, MHeNS, Maastricht University, 6200 MD Maastricht, The Netherlands; 6Institute for Molecular Cardiovascular Research IMCAR, University Hospital RWTH Aachen, 52074 Aachen, Germany; 7Interdisciplinary Centre for Clinical Research IZKF, University Hospital RWTH Aachen, 52074 Aachen, Germany

**Keywords:** placenta, multiphoton microscopy, pre-eclampsia, quantification

## Abstract

Pre-eclampsia is a severe placenta-related complication of pregnancy with limited early diagnostic and therapeutic options. Aetiological knowledge is controversial, and there is no universal consensus on what constitutes the early and late phenotypes of pre-eclampsia. Phenotyping of native placental three-dimensional (3D) morphology offers a novel approach to improve our understanding of the structural placental abnormalities in pre-eclampsia. Healthy and pre-eclamptic placental tissues were imaged with multiphoton microscopy (MPM). Imaging based on inherent signal (collagen, and cytoplasm) and fluorescent staining (nuclei, and blood vessels) enabled the visualization of placental villous tissue with subcellular resolution. Images were analysed with a combination of open source (FIJI, VMTK, Stardist, MATLAB, DBSCAN), and commercially (MATLAB) available software. Trophoblast organization, 3D-villous tree structure, syncytial knots, fibrosis, and 3D-vascular networks were identified as quantifiable imaging targets. Preliminary data indicate increased syncytial knot density with characteristic elongated shape, higher occurrence of paddle-like villous sprouts, abnormal villous volume-to-surface ratio, and decreased vascular density in pre-eclampsia compared to control placentas. The preliminary data presented indicate the potential of quantifying 3D microscopic images for identifying different morphological features and phenotyping pre-eclampsia in placental villous tissue.

## 1. Introduction

Pre-eclampsia (PE) is a systemic vascular disorder affecting 4–5% of pregnant women globally [1]. PE is a leading cause of maternal and foetal morbidity and mortality, and confers a life-long risk to maternal and infant cardiovascular health [2,3,4,5]. Classically, PE manifests as new-onset hypertension and proteinuria after 20 weeks of gestation [6]. Nevertheless, symptoms, severity, time of onset, and outcome can vary significantly, and early diagnosis remains challenging [6,7].

The wide range of clinical phenotypes makes PE more fitting to be a syndrome instead of a single disease. In particular, the distinction of early- and late-onset PE (EO-PE and LO-PE) is frequently applied, and both are likely to have distinct aetiologies [8,9]. Nonetheless, the exact definition of EO-PE and LO-PE is controversial. The current stratification only focusses on time of onset of clinical symptoms, gestational age at time of birth, or a combination of these, without referring to underlying causes [6,9,10].

Quantifying altered placental morphology in different clinical presentations of PE is key to distinguish the clinical syndrome into distinct pathological phenotypes. Currently, 2D histological evaluation is the gold standard for placental investigation [11,12,13], yet none of the histological findings are specific for PE, i.e., are also observed in other placental syndromes such as intra-uterine growth restriction (IUGR) [14,15,16].

More advanced imaging methods such as confocal microscopy have been used for imaging terminal villous vasculature [17,18,19,20]. However, the villous tree is built up of stem villi (of which the diameter can be up to 3000 µm), intermediate villi, and terminal villi [21]. Multiphoton microscopy (MPM) is a tomographic (3D imaging) technique with subcellular resolution. It is based on multiphoton excited fluorescence and features deep tissue penetration (up to 1 mm) [22]. Therefore, thick tissue slabs (containing different components of the villous tree) can be visualized, closely to their native form. Moreover, MPM is efficient in exciting autofluorescence signals and suitable for the label-free imaging of cytoplasm and connective tissue, even in vivo [23].

MPM has been applied to investigate placental membrane architecture [24,25]. However, to our knowledge, the phenotype quantification of the villous tree and vasculature of healthy and PE placentas with MPM has not been described yet. The aim of this study was to develop a MPM placental imaging protocol and to demonstrate quantifiable imaging targets. Quantification of features such as syncytial knots and vasculature characteristics could aid in a more unequivocal characterization of different PE subtypes and improve disease stratification.

## 2. Results

### 2.1. Imaging Targets in Placental Villous Tissue

#### 2.1.1. Trophoblast and Fibrotic Regions: Disorganised Trophoblast in Pre-Eclamptic Placenta

H&E-stained placental chorion illustrated the nuclear accumulations and the villous tree components (Figure 1A,B). As frequently reported in the literature [13,26], EO-PE placenta was characterised by increased syncytial knotting (Figure 1B) in comparison to term control tissue (Figure 1A). As depicted in Figure 1, MPM imaging of unstained tissue (Figure 1C,D and Appendix A) and stained tissue (Figure 1E–H) provided significantly more detailed information about the structural organisation of the trophoblast. Term control placenta was characterized by a structured trophoblastic layer (Figure 1C). Nuclei were closely arranged to each other, and cytoplasmic space was small (inset Figure 1C). The trophoblast of the EO-PE case was disorganised, and numerous nuclear accumulations were present (Figure 1D, Appendix A). Nuclei were unevenly distributed, with larger-appearing cytoplasmic spaces between nuclei (inset Figure 1D). Additionally, excessive collagen accumulation, i.e., fibrosis, was especially observed in the stem and intermediate villi of the EO-PE placenta (Appendix A).

#### 2.1.2. Nuclear 3D Architecture: Syncytial Knots Have Distinct 3D Organisations in PE

Distinct nuclear aggregate shapes on the villi were identified (Figure 1E–H). The term “knots” is often used to describe all nuclear aggregates seen in villi. However, nuclear aggregates can be subdivided into (apoptotic) knots, bridges and sprouts. Knots, mostly found towards term, are condensed nuclear aggregates protruding slightly from the villous surface [27]. Bridges are aggregated nuclei that connect two villi and are thought to provide structural support. Sprouts, mainly common in young placentas, are the initiating points for new developing villi [28]. The sprouts (Figure 1F), bridges (Figure 1G), and knots (Figure 1E,H and Appendix A) could be easily distinguished from one another with 3D information. Prominent nuclear accumulations in term placenta were frequently bridges. Although bridges can be distinguished on 2D slices (Figure 1A), their interpretation is imaging-depth dependent, and they are easily misinterpreted as knots without additional 3D information (Appendix A).

EO-PE placentas were characterized by the presence of wave-like syncytial knots (WLKs) (Figure 1E). Although the WLKs were most dominantly present on stem villi (Figure 1H), this pattern was also observed on intermediate villi (Figure 3A lower red arrow) and in villi from a hypertensive pregnancy (Appendix A). The LO-PE placenta had numerous large, surface-extending knots (Figure 1H). Knots in term control placenta were frequently small, regional, and mildly protruding (Figure 2E). Additionally, knots in term control placenta were observed in close relation with so-called vasculosyncytial membranes (Appendix A), which represent major areas of feto-maternal exchange [29].

#### 2.1.3. Villous Morphology and Vasculature

Villous 3D shapes were highly varying (Figure 3A–D). Depending on achievable imaging depth, linkages of terminal villi, intermediate villi, and stem villi could be visualised. The investigated EO-PE placenta had large paddle-like villous endings (Figure 3A) characteristic of PE placenta. These paddle-terminals were not uncommon and frequently present on villous tips (Appendix A). Term control placenta had shorter but highly condensed villi (Figure 3B), while branching appeared excessive in an IUGR case (Figure 3C). Notable is that the LO-PE placenta had highly varying regions with limited branching (Figure 3D) and highly branched villi (Appendix A). Segmentation procedures of intermediate and terminal villi are illustrated in Appendix A. Visual inspection of images (Figure 3, Appendix A) shows that vasculature morphology mainly follows the villous morphology, especially in terminal villi.

### 2.2. Quantification

#### 2.2.1. Quantification of Nuclei and Their Architecture: MODEL Performance

Nuclei in placental chorion are arranged closely to each other and have varying intensities (i.e., knots are brighter and deeper in tissue, signal-to-noise ratio decreases). This makes intensity-based segmentation very challenging. The machine-based learning approach *Stardist* [30] was employed to segment nuclei. An overview of the developed imaging methodology and quantification is given in Appendix A.

Representative images of placental nuclei of term control and EO-PE are illustrated in Figure 2 and Appendix A. Knots appeared bright and very dense (2D images Figure 2A,E). Corresponding 2D images (Figure 2B,F) and 3D images (Figure 2C,G) segmented with *Stardist* demonstrate the labels of all individual nuclei identified. These images were used to calculate nuclear density. The classification of knots performed with *KnotMiner* is presented in Figure 2D,H.

Although non-significant, EO-PE placentas tended towards higher nuclear density compared to preterm control placenta (Figure 2I). Knot fraction (the villous volume occupied by knots) tended to be higher in EO-PE compared to preterm control placenta (Figure 2J). While significance of LO-PE could not be tested (single value), IUGR knot fraction differed non-significantly from term control. LO-PE and IUGR placenta both tended towards increased knot fraction compared to term control placenta. Notably, preterm placenta did not have a lower knot fraction than term control placenta (Figure 2J).

Knot shape was on average slightly (non-significantly) flatter and (significantly) less elongated in term control and preterm placenta (Figure 2K) compared to IUGR and EO-PE placenta, respectively. Individual knot shape quantification is represented in Appendix A.

#### 2.2.2. 3D Native Villous Tree Organisation: Altered Surface–Volume Relations in Pre-Eclampsia

Quantitative results indicated that villous surface area increased disproportionally to volume in the investigated PE placentas (Figure 3E). Diffusion surface area and villous volume had a positive linear relation in term control placenta, whereas in PE a linear relation between surface area and volume was absent (Figure 3E,F).

The SA/vol ratio provides a measure for the ease of diffusive transport and branching in a biological mechanism to increase the SA/vol ratio. No significant difference was found between the SA/vol ratio of chorionic villi of control and PE or IUGR placenta (Figure 3G). However, the villous SA/vol ratio tended to be highest in LO-PE placenta, which could suggest increased villous branching compared to term placenta (Figure 3G). Notably, the villous SA/vol ratio was lowest in the term IUGR placenta (Figure 3G), while villi appeared highly branched (Figure 3C).

#### 2.2.3. 3D Placental Microvasculature: A Reduced Vascular Fraction in Pre-Eclampsia

For vasculature visualization, staining was performed with UEA-1 lectin [31], which is a direct and less complex method compared to immunostaining [17,18,19,20]. Segmentation was performed with Labkit (Appendix A), and network extraction and quantification with VMTK (Appendix A). The vasculature of term placenta (Figure 4A) appeared highly torturous and branched, whereas LO-PE placenta illustrated a vascular network with longer segments and decreased branching (Figure 4B). Vessel diameters varied throughout a single placental vascular network (Appendix A). Diameters in term control placenta remained steady along a 10 µm equilibrium (Appendix A). In the LO-PE case, diameters varied highly and could become very small (Appendix A).

Vascular fraction measured the percentage of villous volume occupied by vascular structures. Our results demonstrate a lower vascular fraction for the EO-PE placenta compared to (pre-)term control placenta (Figure 4C). Moreover, the vascular fraction in EO-PE was comparable to that of a second-trimester placenta (Figure 4C). LO-PE and EO-PE vascular fraction did not differ significantly, although LO-PE tended towards higher vascular fraction than EO-PE. Vascular content in the LO-PE placenta was significantly lower than in term control placenta when parametric statistical tests were used (Figure 4C).

Network properties were quantitatively investigated by examining branchpoint density (see Appendix A for details on the procedure). Our preliminary data illustrated term placenta vascular networks to have the lowest average branchpoint density (Figure 4D). The highest tortuosity (although not significant) was observed in the IUGR placenta without PE (Figure 4E).

## 3. Discussion

The clinical classification of the main phenotypes of PE, EO-PE and LO-PE, is based on the gestational age. These phenotypes are believed to have different pathophysiologies. Imaging evidence of these differences have been demonstrated with various methods; however, 3D images with subcellular resolution are not as common. In this study, we explored the possibilities of extracting such information with MPM. The goal was to visualize placental tissue with subcellular resolution in 3D, identify imaging targets, and quantify them. We employed MPM because of its ability to image deeper into tissue, and we used a combination of staining and autofluorescence to visualize nuclei, vasculature, fibrillar collagen, and villi. We developed quantification algorithms with the goal of defining quantifiable characteristics that could be used for the classification of different pre-eclampsia phenotypes. Below, we discuss the significance of focusing on the quantification of such features and related challenges.

### 3.1. Trophoblast and Knots

Hypoxia (and associated trophoblast damage) is central in current pathophysiological hypotheses of PE [32]. In PE, trophoblast turnover is no longer balanced. Apoptosis is increased [33] and differentiation and fusion of the so-called cytotrophoblast to form the syncytium is decreased [34], while cytotrophoblast proliferation is reported to be increased or unaltered [35,36,37]. Altered trophoblast function provides a potential clarification for the observed structural abnormalities in PE (Figure 1C,D).

Moreover, placental hypoxia has been found to promote the aggregation of syncytial knots [38]. Increased syncytial knotting is a clear hallmark of pre-eclamptic placentas [13,26]. Distinct 3D conformations (Figure 1E–H) of knots are potentially related to different pathological mechanisms (i.e., being apoptotic or a mechanism to sequester affected nuclei and create regions for diffusional exchange). WLKs in particular were dominantly present on the investigated EO-PE placenta (Figure 1E). Nonetheless, wave-like apoptotic shedding is not specific for PE and has been described in stem villi from cases with severe IUGR without PE [26,39]. The quantification of knots (shape/volume) could potentially aid in the subdivision of pathological, apoptotic, or non-pathological (including sprouts and bridges) knots accordingly. To our knowledge, different shapes of knots have been described [26], but quantitative data regarding relative incidences of alternative knot shapes is lacking. The current developed methodology provides an approach for quantitatively investigating knots and their shapes.

### 3.2. Placental Villous Tree Morphology: Exchange Surface and Volume

Villous branching establishes an increased diffusional surface, and the villous SA/Vol reflects ease of nutrient exchange. Similar to our findings (Figure 3E), a disproportional relationship between surface area and volume has been reported in PE placentas based on a stereological approach [40]. Some stereological estimates only observed altered exchange surface areas in IUGR cases, and not for PE alone [41,42]. Stereology only provides estimates of 3D quantitative parameters, while quantification of MPM images represents the intact 3D villi. A 3D characterization by MPM will therefore aid in more accurate quantification of villous trees to confirm or reject the abovementioned findings.

### 3.3. Placental Villous 3D Microvasculature: Volume and Network Quantification

Terminal villous shapes are influenced by vascular development, and abnormal vascular development is linked to PE [10,43,44]. A vascular casting study demonstrated a reduction of total vasculature in severe PE [43]. Moreover, a higher frequency of avascular villi were described in EO-PE compared to term control [44], providing a potential explanation for reduced vascular fraction (Figure 4C). Moreover, vessel diameters are hypothesized to be influenced by placental hypoxia [10]. Therefore, vessel diameter patterns (Appendix A) provide an interesting imaging target. MPM has enabled the investigation of intact 3D microvascular networks and network quantification. In addition to branchpoint density and tortuosity (Figure 4D,E), network quantification can provide numerous quantitative morphological descriptors, such as vessel length and vessel diameter (Appendix A). According to current hypotheses (Appendix A), LO-PE exhibits increased branching, i.e., predominant vascular development by branching angiogenesis. Conversely, EO-PE placental development is hypothesised to be characterized by a predominance of non-branching angiogenesis [10,45].

During normal development, non-branching angiogenesis is the dominant process of vascular growth later in gestation [46]. Therefore, a younger gestational age is a potential confounding variable, which indicates the importance of including age-matched control placenta to study vascular network properties in PE placenta. Branching differences between EO-PE and pre-term placentas were observed; however, the collection of such samples is more challenging.

The tortuous nature of the placental vessels makes them longer, increases their surface area and slows down blood flow to increase the time for diffusional exchange [18]. High tortuosity, as observed in a IUGR case (Figure 4E), could be an adaptational response for improving diffusion to compensate for low villous SA/vol (Figure 3G). However, excessive tortuosity in vessels is also linked to pathological processes such as hypertension or weakened artery wall stiffness [47]. Vascular twisting would thus be an interesting feature to investigate further regarding placental disease.

### 3.4. Fibrosis

Finally, fibrosis, which is a prominent feature of PE, was identified as a quantitative marker (Appendix A). Since the current analysis focused on intermediate and terminal villi only, fibrosis was not further analysed. Future work focusing on stem villi, where fibrosis was mostly observed, will include the quantification of fibrosis.

### 3.5. Strengths and Limitations

The imaging of the thick tissue (i.e., no need to make thin sections) allowed the imaging of the nearly intact 3D morphology of placental terminal villi. This supported the accurate identification of knots, bridges and sprouts, and allowed for the assessment of the intact 3D organization of the villous tree and intact microvascular networks. Classical histopathology is only able to give a single-plane 2D representation of these complex 3D structures (Figure 1A,B), and therefore, MPM offers a better alternative. Imaging of even larger fields of view by image stitching (2 × 2,5 mm^2^) is possible, as demonstrated in Video S1, although it is more time consuming (3 h).

The MPM penetration depth of unprocessed placenta is limited to approximately 200 μm. Consequently, images of small (and frequently incomplete) networks cause risk for sampling bias. This could be addressed by optically clearing the tissue [48], a process that homogenizes the tissue’s refraction index, to allow imaging down to some millimetres. Combining image stitching and optical clearing would increase the imaged volume several fold. Larger coverage would reduce sampling bias and improve image quantification. However, image acquisition time would increase correspondingly. Acquisition speed could be improved with Light sheet microscopy (LSM) [49]; however, most common LSM microscopes are limited in resolution and penetration depth compared to MPM.

The limited sample size only allowed for the generation of preliminary results, but our goal was to develop and demonstrate the method and its potential. Future efforts should focus on investigating variation between placentas with larger sample size. From our preliminary data, it appears that there are large differences between different phenotypes. This information can be used for substantiated effect size determination. Power calculations using this substantiated effect size could be further used to calculate the number of samples needed to extract statistically significant conclusions. Additionally, obtaining a complete overview of clinical information (including doppler ultrasound findings [50]) is crucial for future research, so that observations can be correlated accordingly. Since we are generating a large set of microscopical quantifiable parameters in combination with the clinical information acquired, it would be advantageous to employ machine learning methods to differentiate between different phenotypes in the future.

## 4. Methods

### 4.1. Tissue Collection and Study Population

A total of 11 placental biopsies were collected from Maastricht University Medical Centre, the Netherlands, between November 2020 and May 2021. All samples were taken from the inner two thirds of the placental disc. Inclusion and exclusion criteria are as follows: PE was diagnosed if blood pressure was ≥140/90 mmHg on two separate occasions accompanied by proteinuria (>300 mg/24 h) after 20 weeks of pregnancy. In absence of proteinuria women, were still eligible in case of multisystem failure. Depending on onset of symptoms (< or >34 w) and time of birth (<or >37 w), placentas were subdivided into early-onset (*n* = 2) and late-onset PE (*n* = 1), respectively. IUGR was suspected if birth weight was below the 10th percentile. Uncomplicated pregnancies were defined as normotensive pregnancies from mothers with no morbidities, without need for additional care, and with delivery ≥ 37 w. Patients with comorbid conditions (e.g., diabetes, asthma) were excluded. C-section and natural birth placenta were both used. A total of 3 placental biopsies from uncomplicated pregnancies, 1 IUGR, and 1 PE sample were collected directly after birth. A total of 6 placental samples (*n* = 2 PE) were collected from the Pathology department of MUMC (archive) and received on formalin. The archived placentas were stored in anatomically correct order after examination, so the inner two thirds of the placental disc could be easily identified and biopsied. Collection and usage of the placentas was approved by the Medical Ethics Committee Academic Hospital Maastricht and Maastricht University (METC, 16-4-047). Clinical characteristics of all placental samples are summarized in Table 1.

### 4.2. Tissue Dissection and Fixation

Placental villous tissue (~20 cm^3^) was sectioned from a central region of the chorion. Samples that were collected directly after birth were stored up to 24 h (depending on the collection time) at 4 °C and frequently washed with phosphate buffered saline (PBS) to remove blood. High blood content can limit imaging depth. Storage did not affect villous and vasculature morphology based on brightfield microscopic observations. Subsequently, it was fixed (1 h) in 4% paraformaldehyde (PFA) and stored in PBS at 4 °C. To increase the number of samples analysed, we also collected formalin fixed samples received from the archive, which were maximally 3 months old. Upon receipt, they were washed 3× with PBS and stored at 4 °C in PBS. Samples from the archive could contain higher blood content, which could limit imaging depth, but imaging more than 150 μm was always possible. Formalin fixation, contrary to organic solvents, does not dehydrate tissue, thus better maintaining the morphology of the tissue. All samples were treated equally in terms of analysis.

### 4.3. Nuclear and Vascular Fluorescent Dyes

Placental villous tissue slabs (~10 mm^3^) were fluorescently labelled to visualize nuclei and vascular structures. Tissue was permeabilized in 0.1% triton, 1% BSA (20 min), and incubated with 10 µg/mL Ulex Europaeus Agglutinin I (UEA I), DyLight^®^ 594 (Vector laboratories, CA, USA, cat#DL-1067-1, em_max_ = 618 nm) for 1–3 h, in the dark at room temperature. After washing with TBS, nuclei were labelled with 5 µM SYTO 13 green (Invitrogen, cat#S7575, em_max_ = 509 nm) for 30 min in the dark at room temperature.

### 4.4. Mounting and MPM Imaging

For imaging, samples were mounted on a 50 mm glass bottom petri dish (MatTek, Ashland, MA, USA) filled with PBS. MPM was performed with a Leica TCS SP 5 (Leica Microsystems, Wetzlar, Germany) multiphoton microscope. Excitation was at 780 nm. Fluorescence was collected with a 20x Leica HCX APO L, NA 1.0 water immersion objective or a 25× NA1.05 Olympus objective. Alternatively, the Leica Stellaris 8 Dive (Leica Microsystems, Wetzlar, Germany) employing similar settings was used.

Autofluorescence and second harmonic generation (SHG) from unstained samples was detected by an external detector with bandpass filter 460–525 nm, and a forward detector with a 380–420 nm bandpass filter, correspondingly. Signal of stained samples was detected using 4 channels. SHG was collected as described before. Internal detectors with detection bandwidths 444–496 nm, 508–559 nm, and 586–650 nm recorded autofluorescence, nuclear stain, and vascular stain, respectively.

All images were acquired in 8-bit in bidirectional scanning mode and at least 1024 × 1024 pixels. Typical field of view was 736 × 736 μm^2^ and imaging depth was around 200 μm and varied between stacks and samples ± 50 μm. Z-step size ranged from 1.5 µm to 5µm. A typical voxel size was 0.7 × 0.7 × 1.5 μm^3^. Image stitching was performed with the Leica Stellaris Dive microscope equipped with motorized stage.

### 4.5. Image processing and Quantitative Evaluation

#### 4.5.1. Segmentation and Volume Quantification of Placental Villous Tissue and Vessels

Images were pre-processed, analysed, and viewed with FIJI [51]. Pre-processing consisted of de-noising with a 3D median filter, sigma 1. Subsequently, placental villi (autofluorescence), vessels, and SHG signals were segmented with the FIJI Labkit plugin [52]. Volume and surface areas of vessels and villi were quantified with 3D ImageJ suite [53] (Appendix A). Binary images were imported into 3D ROImanager [53] to obtain surface area and volume measurements. Villous vascular fraction was calculated by dividing the vascular and villous volume and was converted to percentage. The surface-area-to-volume ratio (SA/Vol) was quantified as a measure for effective diffusion area.

#### 4.5.2. Quantifying Vascular Network Characteristics

Binarized vascular images (exported via FIJI to .nrrd files) were imported into 3Dslicer [54] to quantify vascular network characteristics. The VMTK [55] module of a 3D slicer was used to extract a network model from the vasculature and to provide automatic quantification. More detailed procedures are described in Appendix A.

#### 4.5.3. Nuclear Segmentation and Quantification

Stardist 3D [30] employs a neuronal network to separate densely packed nuclei in 3D image stacks. The model was trained with 5 manually annotated image stacks, and thereafter used to segment placental nuclei.

Quantitative analysis of predicted label maps was performed in FIJI with 3D ROI manager [53]. Nuclear density was defined as the number of nuclei relative to the villous volume (#nuclei/villous volume in mm^3^).

Segmentation of syncytial knots was performed with KnotMiner, available at https://github.com/stegmaierj/KnotMiner (20 December 2022). KnotMiner was built as an extension package for the MATLAB toolbox SciXMiner [56]. The volume of knots was normalised to the volume of the corresponding villus to determine knot fraction. Knot shape was described by elongation index (EI = intermediate principal axis/longest principal axis) and flatness index (FI = short principal axis/intermediate principal axis).

### 4.6. Statistical Analysis

Data are given as mean ±SD. GraphPad prism 5.0 was used for statistical analysis and graph representation. ANOVA with Bonferroni post-test was used to assess significant differences in parameters that were normally distributed. In the case of data that were assumed not normally distributed, Kruskal–Wallis with Dunn’s post-test was employed. If normality could not be statistically determined (by Shapiro–Wilk test) because of too small a sample size, the assumption of normality was made. The *p* values of <0.05 were considered statistically significant. The number images analysed for each case is indicated in Appendix A. All images were treated with the same weight [57,58,59,60,61].

## 5. Conclusions

Our results demonstrated MPM to be a suitable method to visualize placental villous tissue, and the processing methodology developed has the potential to quantifiably differentiate placental morphologies. To understand quantification results, age-matched healthy controls would be necessary. This methodology could be further utilised to phenotype PE subtypes, and ultimately aid in unravelling the multi-etiological nature of pre-eclampsia. In the future, stratification based on 3D placental morphology could aid in the search for early diagnostic markers and (preventative) treatment options to improve maternal and foetal outcomes.

## Figures and Tables

**Figure 1 ijms-24-03240-f001:**
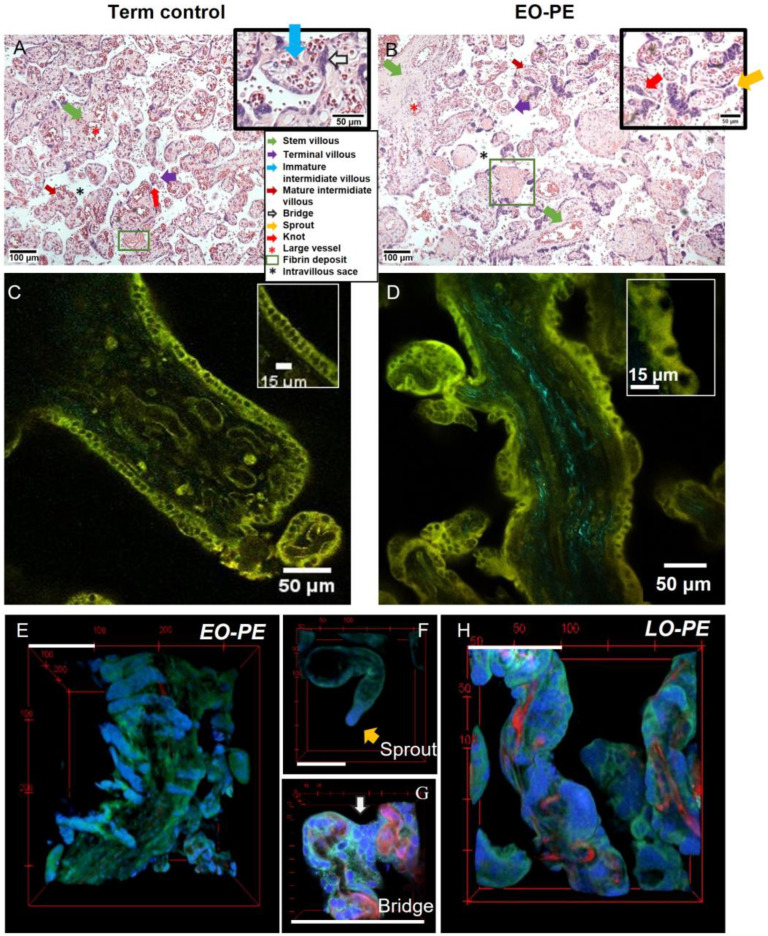
H&E-stained villous tissue of (**A**) term control and (**B**) pre-eclampsia placenta. Villous tissue is visible in pink, and white regions between villi represent the intervillous space. Green arrows point to stem villi (diameter 100–3000 µm) which contain large and centrally located arteries. The blue and dark-red arrow points to an immature intermediate villous or mature intermediate villous, respectively. Purple arrows indicate terminal villi (diameter 30–80 µm). Regions with fibrin depositions are indicated with squares. Nuclear clusters (bridges (white arrow inset (**A**), knots (red arrow) or even sprouts (yellow arrow inset (**B**))) are apparent on term control placenta and excessively present on PE placenta (insets (**A**,**B**)). (**C,D**) Placental tissue imaged with MPM without fluorescent labelling. (**C**) The epithelial trophoblast layer in term control placenta is organised, while (**D**) EO-PE placenta had disorganised epithelial trophoblast. (**E**–**H**) Placental tissue stained for blood vessels (red) and nuclei (blue) to visualise the different 3D shapes of nuclear clusters. (**E**) Wave-like knots on a stem villous from EO-PE placenta. (**F**) The 2nd trimester placenta illustrating a typical sprout and its 3D conformation. (**G**) Nuclear accumulations that form a bridge typically seen in term control placenta. (**H**) LO-PE placenta with various knot shapes, ranging from elongated surface extending shapes to small round protruding knots. Green—villous tissue, blue—nuclei, cyan—collagen, red—vessels. Scale bars in (**E**–**H**)—100 μm.

**Figure 2 ijms-24-03240-f002:**
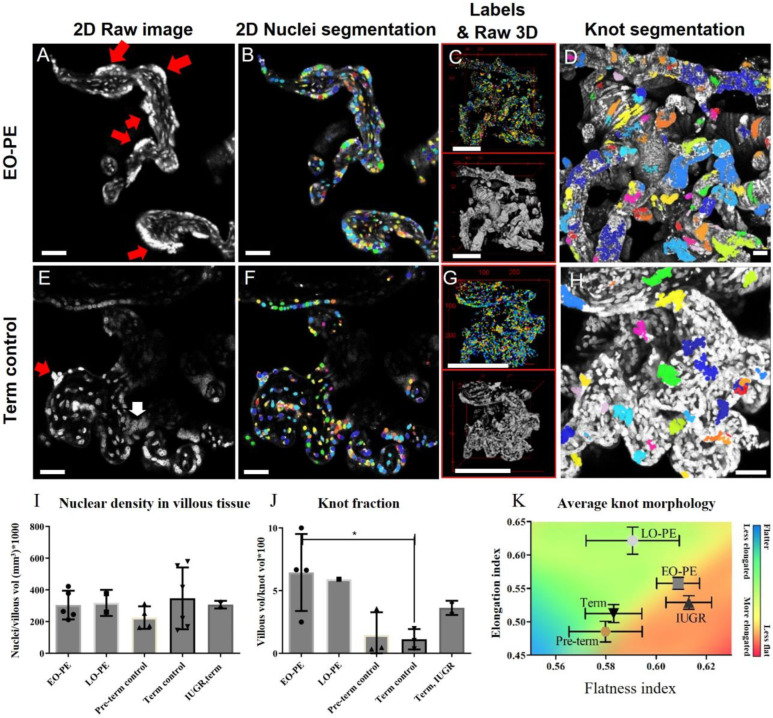
Nuclei segmentation with Stardist and knot segmentation with *KnotMiner*. (**A**) EO-PE, 2D section and (**B**) corresponding segmentation result. Red arrows point to syncytial knots. (**C**) A 3D raw image of nuclei and the corresponding 3D segmentation result. Larger images are available in Appendix A. (**D**) Segmentation of knots with *KnotMiner*, illustrated in the maximum intensity projection of the image. (**E**) A 2D visualisation of nuclei in term control placenta. The white arrow points to a nuclear bridge (can be recognised by dense nuclei connecting two villi, but nuclei are not as bright as compared to knots). (**F**) Segmentation of nuclei in term control placenta. (**G**) The 3D segmentation results of nuclei in term control placenta. (**H**) Knot segmentation in term control placenta by *KnotMiner*. (**I**) Nuclear density (ratio of nuclei over villous volume). EO-PE *n* = 2 placentas (images: 5), LO-PE: *n* = 1 placenta, (images: 2), preterm: *n* = 2 (images 3), term control: *n* = 2 (images 3), IUGR: *n* = 1 (images: 2). (**J**) Knot fraction (percentage of total knot volume over villous volume). (**K**) average knot morphology described by flatness and elongation index. Flatness index was not significantly different for the presented clinical cases. Samples in (J) and (K); EO-PE: *n* = 1 placenta (images:4), LO-PE: *n* = 1 (images: 1), preterm: *n* = 2 (images: 3), term control: *n* = 2 (images: 3), IUGR: *n* = 1 (images: 2). Results are presented as means ± standard deviation. Scale bars in (**A**,**B**,**D**–**F**): 50 μm. Scales bars in (**C**), and (**G**): 200 μm, * *p* < 0.05.

**Figure 3 ijms-24-03240-f003:**
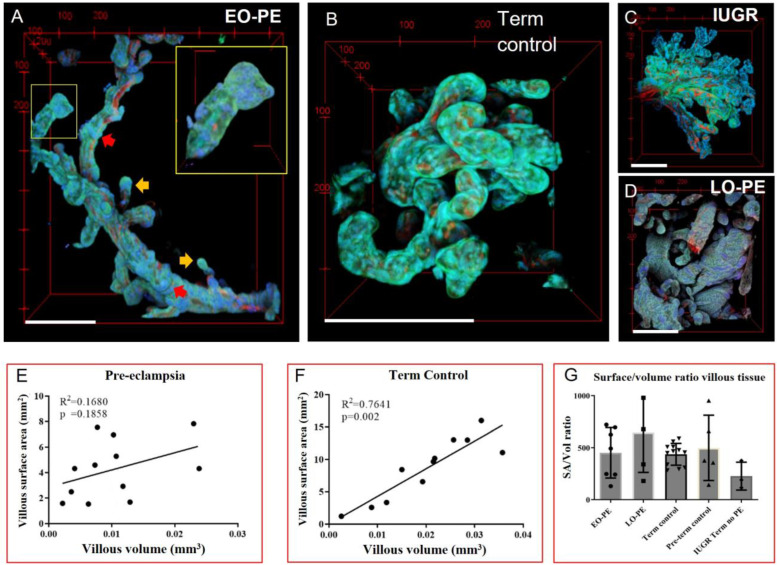
MPM imaging of villous morphology and surface and volume quantification. (**A**) EO-PE placenta had long slender villi. The inset indicates a paddle-like villous ending which was not uncommon for EO-PE. (**B**) Term control terminal villi stem branching off. Villi appear more condensed, more branched and shorter compared to EO-PE. (**C**) IUGR placenta displayed highly branched villi. (**D**) LO-PE illustrating numerous stem villi. (**E**) Villous volume and surface area of pre-eclampsia placenta was quantified and non-linearly related. PE data was pooled together (*n* = 12 images of 2 placentas). (**F**) Villous surface area and volume were linearly related in quantified term control villi (*n* = 11 images, of 3 different placentas). (**G**) Ratio of villous surface area over volume. No clear trend is observed. Green—villous outline, blue—nuclei, red—vascular stain. Red-arrows—(wave-like knots. Yellow arrows—sprouts. Results are presented as means ± standard deviation. Scale bars—200 μm.

**Figure 4 ijms-24-03240-f004:**
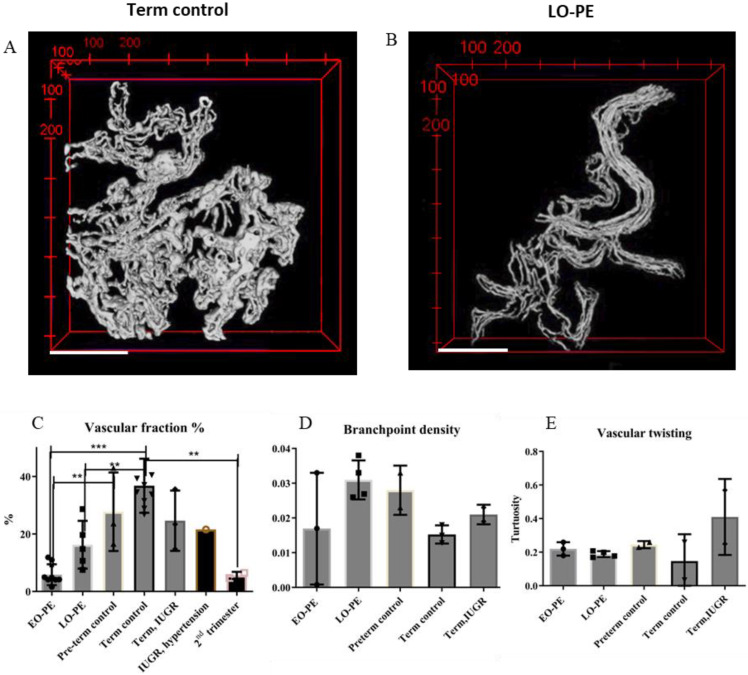
Placental 3D vasculature and network quantification parameters. Segmentation of a connected vascular network from (**A**) term control, (**B**) LO-PE terminal and intermediate villi. (**C**) The vascular fraction was calculated as vessel volume divided by corresponding villous volume of each image. EO-PE *n* = 1 placenta (images: 8), LO-PE: *n* = 1 (images: 5), preterm: *n* = 1 (images: 3), term control: *n* = 3 (images: 9), IUGR: *n* = 1 (images: 3). (**D**) Branchpoint density (#branchpoints/network length (µm)). (**E**) Tortuosity of vessel segments. Tortuosity is defined as the ratio between the length of the curved vessel segment and the line distance in between two branchpoints. Samples in (**D**,**E**); EO-PE *n* = 1 placenta (images: 3), LO-PE: *n* = 1 (images: 4), preterm: *n* = 1 (images: 2), term control: *n* = 2 (images: 3), IUGR: *n* = 1 (images: 2). ** *p* < 0.01, *** *p* < 0.001. Results are presented as means ± standard deviation. Scale bars: 200 μm.

**Table 1 ijms-24-03240-t001:** Clinical characteristics of included patients/placental biopsies. EO-PE—early onset pre-eclampsia, LO-PE—late-onset pre-eclampsia, GA—Gestational age, Gravida/para—number of pregnancies/number of births. SGA/IUGR—Small for Gestational age/intra-uterine growth restriction.

Placenta	Group	Maternal Age (Years)	Gravida/Para	GA	Placental Weight (g)	Birthweight (g)	Collection Method
1	Term control	30	1/0	40 w 2 d	NA	3750	Fresh
2	Term control	38	2/1	40 w 4 d	NA	4040	Fresh
3	Term control	33	2/1	39 w 2 d	NA	3150	Fresh
4	EO-PE	33	1/0	30 w 1 d	190 incomplete	1080	Fresh
5	EO-PE	26	1/0	32 w 3 d	233	1400	Archive
6	LO-PE	26	2/0	37 w 5 d	421	2058	Archive
7	SGA/IUGR	36	3/1	40 w 0 d	NA	3065	Fresh
8	Preterm control	20	2/0	32 w 5 d	328	2089	Archive
9	Preterm control	21	2/1	31 w 5 d	362,5	2134	Archive
10	Maternal hypertension + IUGR	26	1/0	37 w 5 d	282	1895	Archive
11	2nd trimester placenta	21	2/1	22 w 6 d	73	532	Archive

## Data Availability

The data supporting these findings can be found at the Department of Genetics and cell biology, Maastricht University.

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
