# Peer review of "Definition and Quantification of Three-Dimensional Imaging Targets to Phenotype Pre-Eclampsia Subtypes: An Exploratory Study"

_ijms, 2023, doi:10.3390/ijms24043240_

Round 1

Reviewer 1 Report

In this work by Hermans et al entitled “Definition and quantification of three-dimensional imaging targets to phenotype pre-eclampsia subtypes: an exploratory study”, the authors aim to develop a multiphoton microscopy placental imaging protocols for quantification such as syncytial knots and vasculature characteristics. The overall aim here being to offer an early monitoring to recognise different preeclampsia subtypes and improve disease stratification.

The authors analyse overall structures/villous morphology and their volume quantifications as well as nuclei and knot segmentations. They also offer some 3D vasculature and network quantification. The data is overall very strong and they offer some impressive analysis, in relation to the quality of the imaging provided as well as the quantitative measurements offered.

Points to consider:

The key limitation of the study provided is the small number of samples that have been analysed, with only two early-onset and one late-onset pre-eclamptic samples available although the majority of the graphs presented later suggest more data points unless the different values presented came from the same samples. This very small number of conditions significantly lower the impact of the work presented. The quality of the work presented deserves a more representative numbers of samples to clearly determine if their analysis is consistent given the heterogeneity of human samples.

The data provided is very good but why do they not offer examples of late onset pre-eclampic microscopic data in fig 1 and fig. 2?

Reviewer 2 Report

This is a well-presented manuscript with excellent results about the placenta 3D morphology. As the authors commented, the number of samples is insufficient to consider that the study could be comparative or conclusive about the differences among different placental dysfunctions.  However, looking at the state of the art in terms of reports on placental morphology, I consider that the results presented by the authors are valid for publication and will be of interest to those who work in this area. Therefore, I think that the manuscript should be considered for publication.

Minor observations are depicted below.

1. In Page 3, Table 1

The table design is boring and does not provide more information that benefits the results.

Placentas should be grouped and, if possible, the study should contain an equal number of placentas with similar characteristics for each group, even in controls.

The clinical characteristics of the donor women must be included in the table.

2. In Page 6, Figure 1 C and D

The difference in the structure of the trophoblastic layer is remarkable. ¿Have you compared the height of the epithelium and whether there is a difference in this parameter?

3. In Page 8, Figure 2 I, J, and K

I suggest including the (n) for each group.

4. In Page 11, Figure 4 C, D, and E

I suggest including the (n) for each group.

5. In Page 12, Section 4.2

The morphology of the villous tree of the placenta sometimes depends on the biopsy site. In the case of fresh placentas this could be controlled. What certainty is there about the sampling site in the case of archive placentas? Please comment on it.
